# Measurement of China's green development level and its spatial differentiation in the context of carbon neutrality

Kun Liang[1]*, Li Luo[2]

1 School of Economics, Sichuan University, Chengdu, Sichuan, China, 2 Sichuan Engineering Technical College, Deyang, Sichuan, China

☯ These authors contributed equally to this work.
* 724170246@qq.com

**Data Availability Statement:** All relevant data are within the manuscript and its Supporting Information files.

**Funding:** This research was supported by the General Project of Chinese Medicine Culture

## Abstract

China's green development in the context of carbon neutrality is facing both new constraints and new practices. In the new situation, it is crucial to further refine the indicator system and re-measure green development. Based on previous studies and carbon neutral background, the theoretical framework of green development is improved and an indicator system is constructed, and the panel entropy method, Thiel index and Moran index are used to measure and analyze green development level and its spatial evolution pattern in China from 2010 to 2019. The study shows that: (1) China's green development level is on the rise and its growth rate is high, and it generally shows a spatial distribution pattern of northeast >east > west > central, and the fastest growth rate in the east; (2) There are some spatial differences in green development level in China and the spatial differences are gradually narrowing, while only the spatial differences in the green wealth index and the inter-regional differences in the green growth index are slightly expanding; (3) There is no obvious spatial clustering phenomenon in China's green development level as a whole, and there is a positive spatial matching relationship between green development level and green wealth index and green growth index locally, which does not form a close correlation with green welfare index. The study proposes countermeasures based on stimulating the endogenous power of green development and narrowing the regional differences of green growth index, which are of reference value for China to promote green development in the context of carbon neutrality.

## Introduction

Regional green development faces carbon neutrality issues of great concern to the world. Countries are actively looking for effective paths and means to reduce carbon emissions, for example, the United States is investing heavily in clean energy, Japan has proposed a national strategic goal of a "low-carbon society", and South Korea has selected 39 key carbon neutral technologies to focus on development [1, 2]. The EU has been a leader in the global green development and sustainability process. In 2019, the *European Green Deal*, officially released by the European Commission, sets out the main approaches to reducing greenhouse gas

Heritage and Research Center of Sichuan Province, China (grant no. 2020Y006); Zhoujing received the award, her website URL is: https://whccyyjzx. scctcm.edu.cn/info/1016/1102.htm; The funders had no role in data collection and analysis, preparation of the manuscript.

**Competing interests:** The authors have declared that no competing interests exist.

emissions. The EU has integrated sustainability factors into all policy formulation, continuously promoted the construction of a sustainable investment and financing system, formulated national environmental budgets, accelerated the construction of a carbon emission trading market, attached importance to green and digital technology innovation, increased the transformation of clean energy, promoted the transformation of industry to clean and circular, accelerated the transformation to sustainable and smart mobility, attached great importance to ecosystem protection and restoration, sustainable use of resources, and human health improvement [3]. As of March 2021, the Paris Agreement, which has 195 signatories, seeks to hold the global average temperature increase to less than 2° C above pre-industrial levels, with more countries taking key initiatives to achieve carbon neutrality in the future. In 2021, China's 14th Five-Year Plan proposed to "strive to achieve carbon neutrality by 2060", setting a clear target for China's carbon emission reduction tasks and putting forward higher requirements for green development nationwide [4]. Through green development approaches such as green low-carbon technology innovation [5], renewable energy development [6], green industry cultivation [7, 8], green lifestyle promotion [9], ecosystem restoration and ecological function restoration [10, 11], the region promotes and supports carbon emission reduction in various economic and social fields. Therefore, the existing evaluation indicator system of green development level should be adjusted according to the realistic needs of carbon neutrality, which needs to include both carbon emission constraint indicators and indicators representing new trends or key areas of green development, in order to better understand the stage state of regional green development and its characteristics in the new context.

Green development is based on the holistic and symbiotic nature of the system, emphasizing the joint green development of ecological, economic, and social systems, thus enhancing economic vitality, social well-being, and nature resource wealth [12]. In existing studies, there are several different views on the understanding of green development level: The first view is that green development is a process of high-quality development, and thus green development level is a quantification of the pressure, state, and response in this process [13]; The second view is that green development level is considered in terms of process and outcome, which involves environmental governance, natural resource utilization and environmental quality [14]; The third view is that the level of green development represents a quality of development, which involves the quality of the environment, the quality of ecosystems and the level of production and living [15]. According to the Oxford Advanced Learner's Dictionary, "level" indicates the state or result of the stage in which the object of concern is located. Therefore, the level of green development indicates the result of green development of economic-social-ecological system at a certain stage.

The construction of green development level indicator system is the key to the quantitative research on green development. Early studies on green development level indicator system mainly focus on two major aspects: economic growth and resource and environmental constraints. A more representative one is the Green Development Index proposed by Beijing Normal University in 2010, which contains the greenness of economic growth, resource and environmental carrying potential, and government policy support, and was subsequently applied to the evaluation of different regions [16, 17]. In 2016, the Green Development Indicator System released by the National Development and Reform Commission of China mainly involves several aspects of resources, environment, ecology, growth and life. The aforementioned indicators pay little attention to the social dimension and do not fully reflect the basic elements in the connotation of green development. Subsequent studies in this area have supplemented the indicators of social dimensions of green development, such as adding social progress dimensions of population, social green development, human development, and social equity [18–22]. As the research progressed, scholars constructed the indicator system of green

development level based on different theories. Firstly, the indicator system is constructed from four dimensions of green ecology, green life, green production and green policy [15, 23], as well as indicators from five dimensions of resource utilization, industrial greening, economic development quality, environmental protection and green habitat [24–26], in combination with the ecology-production-living space theory. The ecology-production-living space theory can intuitively explain the relationship between the roles of production, life and ecology in green development practice scenarios. Secondly, based on the pressure-state-response (PSR) model and the DPSIR model, the whole process of green development is measured from economic state, environmental pressure, social and policy response or environmental response [27, 28], but these evaluation indicator systems ignore the part of natural resources. Thirdly, based on the three subsystems of ecology, economy and society, the evaluation system is constructed from social green development, economic green development, resource green development and environmental green development [20]. Fourthly, based on the three-loop model of green development, the indicator system is designed from green growth, green welfare, and green wealth [29, 30]. This model better explains the mechanism of the positive effect of the nature-economy-society complex system in green development, but it should be noted that some connotations of green wealth and green welfare are subject to discretion, for example, green wealth includes social capital, so the connotations of the three dimensions of this model need to be adjusted and revised before they can be used for the construction of green development level indicators. Therefore, the connotations of the three dimensions of this model need to be adjusted and revised before they can be used in the construction of green development level indicators. In the context of today's global carbon emission reduction, carbon dioxide emissions and new energy generation are included in the green development level index system [22, 31]. At present, the green development level evaluation study has involved countries along the belt and road, ASEAN region, provinces in China, China's Yangtze River Economic Belt, resource-based cities and other regions [20, 22, 23, 31]. As for the methods of green development level evaluation, there are entropy method, TOPSIS model, improved TOPSIS model, entropy power TOPSIS model, projection tracing evaluation model, principal component analysis method, PSR model, and longitudinal and S-type cloud model [26, 27, 32, 33]. Follow-up studies need to carefully select the methods that can evaluate the level of green development from the perspective of "results".

The measurement of green development level at home and abroad is gradually deepening, and there is room for further expansion. Firstly, the theoretical framework of green development is still immature and needs to be further improved based on the development of practice and carbon neutral background in order to be used as the theoretical support for the indicator system of green development level. Secondly, the existing indicator system emphasizes the stock of natural resources and "passive defense" environmental protection and policy support, but does not reflect the highlights and advantages of the ecosystem in green development and the output of the ecosystem. Third, the existing indicator system and its economic part are still lacking in highlighting green industries, and lacking the indicators of carbon emission constraints. Based on the above analysis, the study takes 30 Chinese provinces from 2010 to 2019 as samples and focuses on the following aspects: firstly, based on the carbon neutral background and the new practice of ecological civilization construction, the theoretical framework of green development is further improved, and the basic framework of the indicator system of green development level is constructed on this basis; secondly, the indicators related to ecological service function, green industry development, carbon emission constraint and social welfare are incorporated into the indicator system of green development level; thirdly, the analysis of spatial evolution characteristics is carried out from the green development level and its three constituent index to answer the

questions of regional characteristics, regional differences and spatial clustering of China's green development level, and targeted suggestions are made according to the spatial differentiation characteristics.

## Construction and methodology of green development level indicator system in the context of carbon neutrality

### Rationale of the indicator system

The green development level indicator system needs to be based on the theoretical framework of green development. Therefore, based on previous studies [24, 29, 34], this study draws on the perspectives of ecological modernization, market environmentalism and environmental economics [35], and further improves the theoretical framework of green development according to the three major systems of nature, economy and society by combining the carbon neutral context with the practice of ecological civilization construction in China (Fig 1). The theoretical framework of green development in the context of carbon neutrality is composed of three major systems, namely, nature, economy and society, which interact positively around the target layer, connotation layer and benchmark layer. Specifically: Firstly, it is the baseline layer. This layer is composed of natural system, economic system and social system. Secondly, it is the inner layer. This layer is based on green wealth, green growth as a means and green welfare as a goal, and promotes the positive interaction among the three systems of nature, economy and society, forming the inner layer of green development. To expand: (1) green wealth is based on various natural elements, involving not only the survival and protection of natural resources, but also supply services, regulation services, ecotourism services and other ecosystem service functions [36], especially $CO_2$ absorption function. (2) Green growth mainly includes three parts: economic growth, ecological economy and green industry. Among them, ecological economy is mainly to reduce energy consumption and undesired output in economic activities, such as reducing pollutants that directly damage the ecological environment and greenhouse gases that affect global climate change; green industry is mainly based on renewable energy, green innovative technologies and other green elements to form new green and low-carbon industries. (3) Green welfare includes both the configuration of green and low-carbon facilities, equipment, environment, and services in the public space and daily life space of society, as well as green and low-carbon consumption in social life, while emphasizing the increase of people's well-being [37]. Thirdly, it is the target layer. In terms of sub-goals, the natural system achieves surplus ecological assets and sustainable ecological functions, the economic system achieves green growth, and the social system achieves green living,

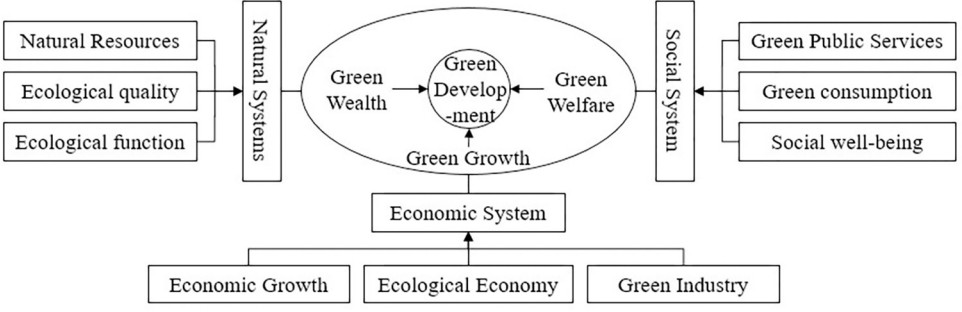

**Fig 1. The theoretical framework of green development theory.**

comprehensive human development, social harmony, and other well-being. From the overall goal, the whole nature-economy-society complex system achieves green development with the positive effect of each element in the baseline layer and the continuous accumulation of each dimension in the inner layer. This paper constructs the corresponding indicator system based on the above theoretical framework.

## Details of the composition of the indicator system

According to the theoretical framework of the "three layers" of green development, the green development indicator system is designed according to the data availability and practicality, with reference to relevant research results at home and abroad and the top-level national design. In principle, the selection of indicators emphasizes the fairness of regional comparison, so most of the indicators are proportional, average (per capita or per land) and intensity. This study established a green development level indicator system covering 3 system layers, 9 element layers and 37 indicators (Table 1).

## Green development level indicator system measurement method

In this paper, the entropy value method is used to calculate the weights and results of each green development level. Firstly, the data of the indicators are standardized. The original data of positive and negative indicators of green development level are standardized according to Eqs (1) and (2). Among them, the set of regional green development level indicators is $\{X^*_{ijr}|$ $i = 1,2,...,m; j = 1,2,...,n; r = 1,2,...,p\}$, $X_{ijr}$ is the value of the $r$th indicator of the green development level of province $j$ in year $i$, $min(X_r)$ and $max(X_r)$ denote the minimum and maximum values of the $r$th indicator, respectively, $X^*_{ijr}$ is the standardized index value of $X_{ijr}$.

$$\text{Positive indicators}: \quad X^*_{ijr} = [X_{ijr} - min(X_r)]/[max(X_r) - min(X_r)] \tag{1}$$

$$\text{Negative indicators}: \quad X^*_{ijr} = [max(X_r) - X_{ijr}]/[max(X_r) - min(X_r)] \tag{2}$$

Secondly, the entropy method formula of panel data [27] was applied to calculate the level of green development at each provincial level in China for each year.

Weight of indicator $r$ in region $j$ in year $i$:

$$Y_{ijr} = X^*_{ijr}/\sum_{i=1,j=1}^{m,n} X^*_{ijr} \tag{3}$$

Information entropy of indicator $r$:

$$E_r = -\frac{1}{ln(mn)} \cdot \sum_{i=1,j=1}^{m,n} (Y_{ijr} \times lnY_{ijr}) \tag{4}$$

$$\text{Weighting of indicator } r: \quad W_r = (1 - E_r)/\sum_{r=1}^{p} (1 - E_r) \tag{5}$$

Composite level score of region $j$ in year $i$:

$$S_{ij} = \sum_{r=1}^{p} (W_r \times X^*_{ijr}) \tag{6}$$

**Table 1. Evaluation indicator system of China's green development level in the context of carbon neutrality.**

| System Layer | Element Layer | Indicator Layer | Properties | Weights |
|---|---|---|---|---|
| A. Green Wealth (0.4190) | $A_1$ Resource Abundance (0.4197) | $A_{11}$ Per capita water resources ($m^3$/person) | + | 0.3883 |
| | | $A_{12}$ Arable land area per capita ($hm^2$/person) | + | 0.2571 |
| | | $A_{13}$ Forest area per capita ($hm^2$/person) | + | 0.3546 |
| | $A_2$ Ecological quality (0.3531) | $A_{21}$ Forest cover (%) | + | 0.1994 |
| | | $A_{22}$ Wetland area as a proportion of national land area (%) | + | 0.5583 |
| | | $A_{23}$ Nature reserve area as a proportion of the jurisdictional area (%) | + | 0.2423 |
| | $A_3$ Ecological service function (0.2272) | $A_{31}$ Food production per capita (t/person) | + | 0.1695 |
| | | $A_{32}$ Soil conservation value to national land area ratio (Yuan/ $hm^2$) [a] | + | 0.0787 |
| | | $A_{33}$ Ratio of flood water storage value to national land area (Yuan/ $hm^2$) [a] | + | 0.2482 |
| | | $A_{34}$ Ratio of the value of purifying the environment to the area of the country (Yuan/ $hm^2$) [a] | + | 0.0526 |
| | | $A_{35}$ Carbon sequestration and oxygen release value to national land area ratio (Yuan/ $hm^2$) [ab] | + | 0.0916 |
| | | $A_{36}$ Interference regulation value to national land area ratio (Yuan/ $hm^2$) [ac] | + | 0.2616 |
| | | $A_{37}$ Number of national forest parks (division) | + | 0.0977 |
| B. Green Growth (0.4455) | $B_1$ Economic Growth (0.5012) | $B_{11}$ GDP per capita ($10^4$ Yuan/person) | + | 0.3371 |
| | | $B_{12}$ Local fiscal revenue per capita (Yuan/person) | + | 0.6629 |
| | $B_2$ Ecological Economy (0.0616) | $B_{21}$ Fertilizer (converted) application per unit of arable land area (t/$km^2$) | - | 0.2881 |
| | | $B_{22}$ COD emissions per unit GDP of wastewater (t/$10^4$ Yuan) | - | 0.1778 |
| | | $B_{23}$ $SO_2$ emissions per unit GDP (t/$10^4$ Yuan) | - | 0.0972 |
| | | $B_{24}$ Energy consumption per unit of GDP (t standard coal/$10^4$ Yuan) | - | 0.2759 |
| | | $B_{25}$ $CO_2$ emissions per unit of GDP (t/$10^8$ Yuan) [d] | - | 0.1609 |
| | $B_3$ Green Industry (0.4371) | $B_{31}$ Green Total Factor Productivity (%) [e] | + | 0.1735 |
| | | $B_{32}$ Wind, light and water power generation as a proportion of total power generation (%) | + | 0.2416 |
| | | $B_{33}$ Number of green invention applications (pieces) | + | 0.3749 |
| | | $B_{34}$ Agricultural output value per unit of arable land area ($10^4$ Yuan /$km^2$) | + | 0.1273 |
| | | $B_{35}$ Percentage of added value of tertiary industry (%) | + | 0.0825 |
| C. Green Welfare (0.1355) | $C_1$ Green Public Services (0.2419) | $C_{11}$ Number of public toilets per million people in cities (seats per $10^4$ people) | + | 0.3536 |
| | | $C_{12}$ Urban sewage treatment rate (%) | + | 0.0561 |
| | | $C_{13}$ Harmless treatment rate of urban domestic waste (%) | + | 0.1028 |
| | | $C_{14}$ Cities have public trams per million people (standard units / $10^4$ people) | + | 0.3579 |
| | | $C_{15}$ Greening coverage of built-up areas (%) | + | 0.1295 |
| | $C_2$ Green consumption (0.3245) | $C_{21}$ Per capita energy consumption (kg standard coal/person) | - | 0.3274 |
| | | $C_{22}$ Per capita domestic electricity consumption (kW-h/person) | - | 0.1278 |
| | | $C_{23}$ Per capita daily domestic water consumption (L/person) | - | 0.5448 |
| | $C_3$ Social well-being (0.4030) | $C_{31}$ Engel coefficient of rural residents (%) | - | 0.3758 |
| | | $C_{32}$ Disposable income ratio between urban and rural areas (%) | - | 0.1294 |
| | | $C_{33}$ Years of education per capita (years) | + | 0.2833 |
| | | $C_{34}$ Number of health facility personnel per million people (persons) | + | 0.2116 |

[a] The calculation formulas refer to the literature [38, 39].

[b] The average value of NPP in the calculation of carbon sequestration and oxygen release value is the province average data of NPP.

[c] The average return of natural output of food in the calculation of disturbance regulation value is 2 Yuan/$m^2$.

[d] $CO_2$ emissions in the indicator system are calculated from energy consumption-related data, and $CO_2$ emissions generated by cement production are not included.

[e] B31 is measured using super-efficient SBM method. The desired output of Green Total Factor Productivity is GDP, the undesired outputs are wastewater's COD emissions, $SO_2$ emissions, and $CO_2$ emissions, and the input indicators are labor (number of urban units employed at the end of the year), capital (amount of investment in fixed assets), and energy [40]. GDP is deflated by using 2004 as the base period and the amount of fixed asset investment is deflated by using 2004 as the base period for the price index.

## Measurement of China's green development level in the context of carbon neutrality

### Data sources

In this paper, a sample of 30 provinces (including autonomous regions and municipalities directly under the Central Government) in China, excluding Tibet Autonomous Region, Taiwan Province, Hong Kong and Macao Special Administrative Regions, was selected for this study. The data were mainly obtained from China Statistical Yearbook [41], China Energy Statistical Yearbook [42], China Electricity Yearbook [43], China Environmental Statistical Yearbook [44], China Forestry and Grassland Statistical Yearbook [45], China Labor Statistical Yearbook [46], as well as provincial yearbooks [47], China Carbon Accounting Database [48], and the Application Service Platform of Land Survey Results of the Ministry of Natural Resources of China [49]. In addition, the data of 500-m vegetation net primary productivity mean value were obtained from the MOD17A3HGF.006 datasets from NASA website [50]. The missing data of individual indicators are treated as follows: for the missing data in the middle years, the average value method is mainly used to supplement them; for the missing data in the first and last years and the indicators that have changed slightly only in a long time, the method of using the nearest year is mainly used to supplement them.

### Characteristics of China's green development level and its composition index

According to (1) to (6), China's green development level and its composition index from 2010 to 2019 are measured, and the results are shown in Fig 2.

According to the measurement results, China's green development level under the carbon neutral background shows the following characteristics: firstly, the green development level is low, and the maximum value is 0.3848 in 2019, which indicates that China's green development still has more room for improvement. For this reason, China's 14th Five-Year Plan in 2021 proposes to build an ecological civilization system, promote a comprehensive green transformation of economic and social development, and formally establish the carbon peaking and carbon neutral targets, which means that China will continue to expand the depth and breadth of green development. Secondly, the overall green development level shows a rapid growth trend, with the average value increasing from 0.2469 in 2010 to 0.3848 in 2019, with an average annual growth

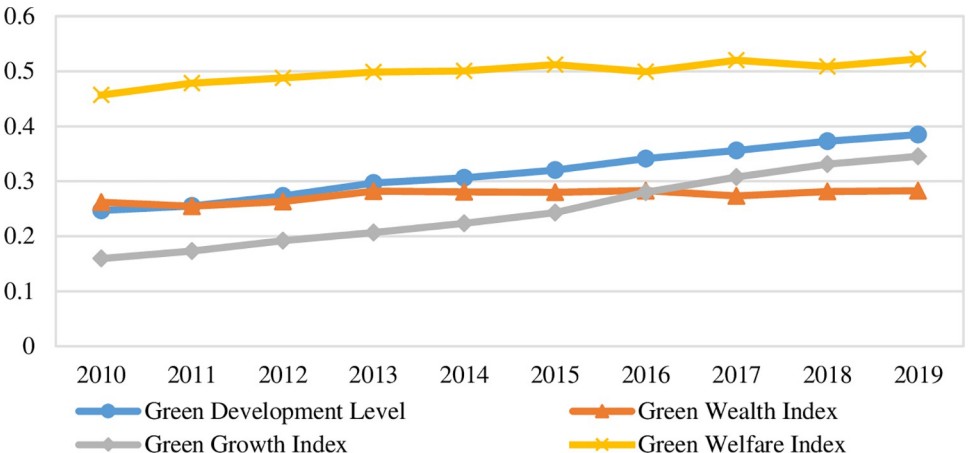

**Fig 2. Trends of China's green development level and its composition from 2010 to 2019.**

rate of 5.05%, because the 18th National Congress of the Communist Party of China in 2012 proposed to vigorously promote the construction of ecological civilization, and the 5th Plenary Session of the 18th Central Committee in 2015 made green development as one of the five major development concepts, and the central government comprehensively deployed green development strategy, local governments comprehensively implement the concept of green development, making the level of green development can be rapidly improved.

From the results of the three composition index of green development measurement: firstly, the green wealth index has a slow upward trend, from 0.2621 in 2010 to 0.2829 in 2019, with an average annual growth rate of 0.85%; secondly, the green growth index shows a rapid upward trend, from 0.1594 in 2010 to 0.3454 in 2019, with an average annual growth rate of 8.97%; thirdly, the green welfare index fluctuates slightly, but shows a steady upward trend overall, growing from 0.4567 in 2010 to 0.5223 in 2019, with an average annual growth rate of 1.50%. Overall, all three composition index show an upward trend, with the green growth index showing the largest increase, thanks to the significant increase in GDP per capita and fiscal revenue per capita as well as the rapid development of green industries. The green welfare index is the largest among the three composition index, which indicates that China's green public service facilities and equipment are improving, while green consumption is being promoted and social welfare is gradually improving. Therefore, in the context of carbon neutrality, China needs to continue to accelerate the green transformation of industries, especially to speed up the replacement of traditional energy by new energy sources and the research and development of green technologies.

## Characteristics of China's regional green development level and its composition index

In order to analyze the regional green development, this paper divides the 30 provinces into four regions: eastern, central, western and northeastern, and analyzes the level of green development and its composition index in the four regions, as shown in Fig 3.

1. Characteristics of China's regional green development levels. Fig 3A shows the green development levels of the four major regions and their development trends. In terms of changing trends, the green development levels of the four major regions show an overall increasing trend, which means that the green development levels of the four major regions in China are constantly improving. From the average value, the green development shows a spatial pattern of the Northeast (0.3945) > East (0.3660) > West (0.2886) > Central (0.2402), where Northeast and East are significantly ahead of the national average (0.3153), while West and Central are below the national average and the difference between these two regions tends to converge. In terms of the average annual growth rate, the average annual growth rate of green development level is higher, from high to low, in the order of the East (5.93%), Central (5.78%), West (4.74%) and Northeast (2.39%), where there is a trend of slowing down in the green development rate in northeast region. It can be seen that the gap between the eastern region and the central and western regions is increasing and the gap with the northeast region is decreasing.

2. Characteristics of each composition index of regional green development in China. In order to further analyze the reasons of the differences in green development levels across regions, this paper compares and analyzes each composition index by region separately, as shown in Fig 3B–3D. From the average value, the three composition index of green development in the four major regions show fluctuating growth, among which the green wealth index of the northeast region (0.4507) is significantly higher than that of the West (0.2826), East (0.2463)

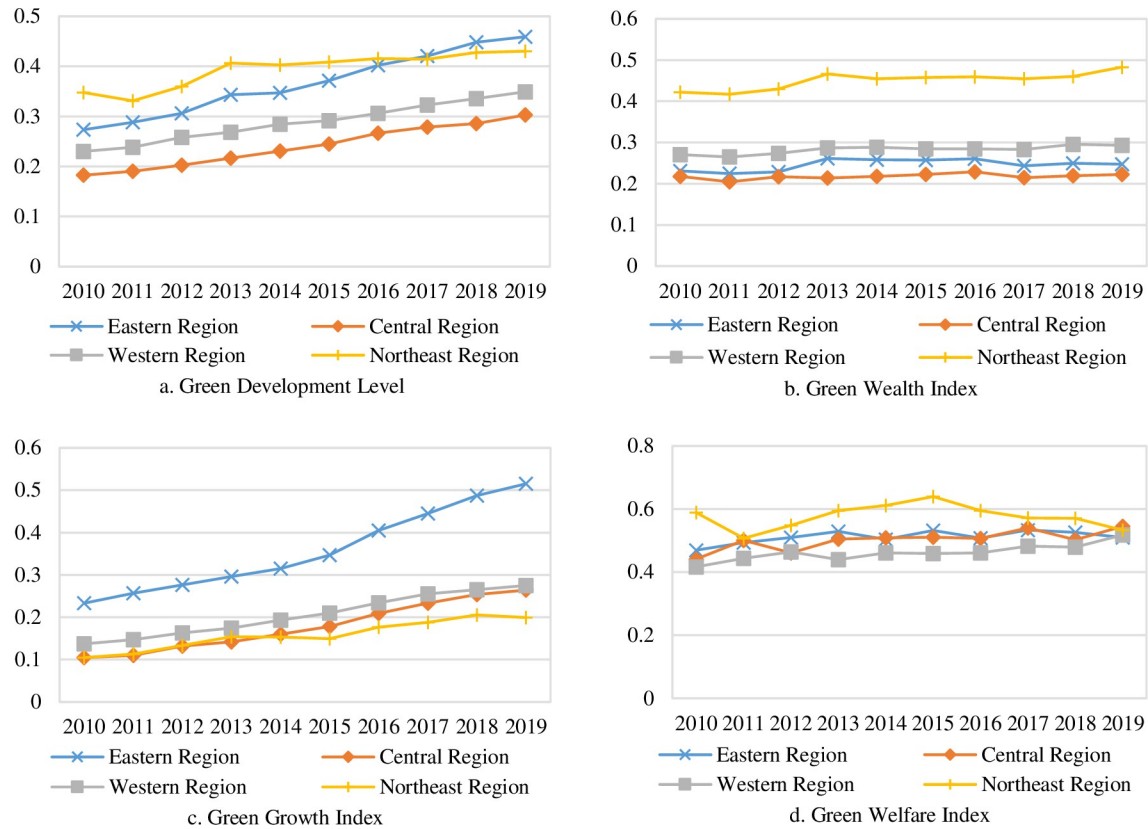

**Fig 3. Trends of green development level and its composition in four major regions of China from 2010 to 2019.**

and Central (0.2179), and the green growth index of the east region (0.3577) is significantly higher than that of the West (0.2054), Central (0.1787) and Northeast (0.1578), and the regional differences in the green welfare index are smaller. In terms of growth rate, the highest growth rate of green wealth index is in the Northeast (1.50%) and the lowest is in the Central (0.22%); the highest growth rate of green growth index is in the Central (10.84%) and the lowest is in the Northeast (7.44%); the highest growth rate of green welfare index is in the West (2.45%) and the lowest is in the Northeast (-1.13%). Among them, although the Northeast is rich in natural resources and has a good manufacturing base, the lagging institutional innovation, high proportion of state-owned enterprises, and serious population loss cause slow economic growth, and insufficient application of green technology causes environmental pollution, all of which inhibit the potential for green economic and social development in the Northeast. In a comprehensive view, the green wealth index shows a spatial pattern of Northeast > West > East > Central, while the green growth index shows a spatial pattern of East > West > Central > Northeast, which shows that the region with more green wealth do not turn the "green mountains" into "golden mountains" better.

## Characteristics of green development level and its composition index by provinces in China

China has taken the initiative to reduce carbon emissions since the 12th Five-Year Plan, and since the carbon neutrality target will only be formally set in China's 14th Five-Year Plan in

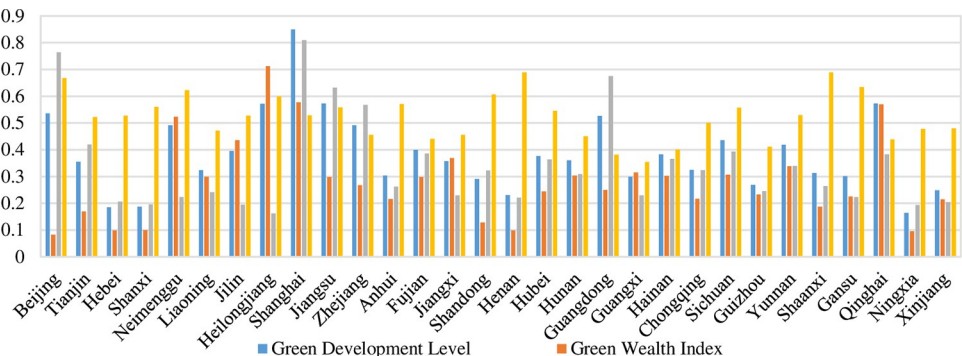

**Fig 4. China's green development level and its composition in 2019.**

2021, this paper further analyzes the differences in the development of green development levels and its component index among 30 provinces, taking 2019 as an example, as shown in Fig 4.

1. The characteristics of green development level of each province in China. In 2019, the average value of China's green development water is 0.3848, with 12 provinces above the average, accounting for 40% of the country's provinces, including 6 provinces in the eastern region, 4 provinces in the western region, and 2 provinces in the northeastern region, specifically Beijing, Inner Mongolia, Jilin, Heilongjiang, Shanghai, Jiangsu, Zhejiang, Fujian, Guangdong, Sichuan, Yunnan, and Qinghai. The main reasons why these provinces have higher than average green development levels vary: the main reason for Shanghai, Jiangsu and Sichuan is that the three index of green wealth, green growth and green welfare are all higher, Beijing is mainly influenced by green growth and green welfare, Inner Mongolia, Jilin, Yunnan and Heilongjiang are mainly influenced by green wealth index and green welfare index, and Fujian and Qinghai is mainly due to higher green wealth index and green growth index. There are a total of 18 provinces below the national average, accounting for 60% of the national provinces, including 4 provinces in the eastern region, 6 provinces in the central region, 7 provinces in the western region, and 1 province in the northeast region, including Tianjin, Hebei, Liaoning, Shandong, Hainan, Shanxi, Anhui, Jiangxi, Henan, Hunan, Hubei, Guangxi, Chongqing, Guizhou, Shaanxi, Gansu, Ningxia, and Xinjiang. It can be seen that these provinces are mainly concentrated in the central and western regions.

2. The characteristics of China's regional green development composition index. In 2019, the average value of China's green wealth index is 0.2829, above the average value of 14 provinces, and concentrated in the Northeast and developed areas of the eastern coast, some provinces in the West, these provinces are rich in natural resources and contribute significantly to ecological conservation, performing important ecological functions, thus becoming natural treasures with superior natural landscapes and good ecological quality in China. The average value of the national green growth index is 0.3454, and the 11 provinces above the average value are Beijing, Tianjin, Shanghai, Jiangsu, Zhejiang, Fujian, Shandong, Hubei, Guangdong, Hainan, Sichuan and Qinghai, and they are concentrated in the eastern region, which are the leaders of green system and technological innovation and have obvious economic advantages, thus promoting the faster development of green industries and reducing environmental pollution and carbon emission intensity. The average value of the national green welfare index is 0.5223, with 16 provinces above the average value. These provinces have continuously optimized urban green infrastructure, adequate public health

and environmental protection facilities and equipment, relatively well-constructed low-carbon transportation systems, and gradually deepened green and low-carbon lifestyles for all people, thus promoting the overall improvement of green welfare.

## Analysis of regional differences in China's green development level in the context of carbon neutrality

### Thiel index

In this paper, the provinces involved in the study are divided into four groups ($z = 1,2,3,4$) in the East, Central, West, and Northeast, and the overall gap $T$ in China's provincial green development level and the between-group gap $T_B$ and within-group gap $T_W$ in each of the four groups are calculated by applying the Thiel index Eq (7) to (9). $X_Z$ is the proportion of the green development level of group $Z$ to the total green development level of all provinces in China, and $X_j$ is the proportion of the green development level of province $j$ to the total green development level of all provinces in China. In addition, $n$ is the total number of provinces involved in the study, $n_Z$ is the number of provinces in group $z$, and $g_Z$ is the group of provinces. The between-group gap $T_B$ and within-group gap $T_W$ are the two basic sources that form the overall gap $T$.

$$T = T_B + T_W \tag{7}$$

$$T_B = \sum_{z=1}^{7} [X_z ln(X_z n/n_z)] \tag{8}$$

$$T_W = \sum_{z=1}^{7} \{X_z \sum_{i \in g_z} [(X_j/X_z)ln(X_j n_z/X_z)]\} \tag{9}$$

### Regional variation analysis

To continue the study of the overall regional differences and the sources of regional differences in the level of green development among provinces in China from 2010 to 2019, the Thiel index was calculated based on Eqs (7) to (9) and decomposed into within-group gaps and between-group gaps, as shown in Table 2.

**Table 2. Results of China's green-development level and its composition index from 2010 to 2019 applying Thiel index.**

| Type | | 2010 | 2011 | 2012 | 2013 | 2014 | 2015 | 2016 | 2017 | 2018 | 2019 |
|---|---|---|---|---|---|---|---|---|---|---|---|
| Green Development | T | 0.0891 | 0.0827 | 0.0693 | 0.0754 | 0.0689 | 0.0668 | 0.0668 | 0.0650 | 0.0671 | 0.0660 |
| | $T_B$ | 0.0176 | 0.0146 | 0.0148 | 0.0199 | 0.0149 | 0.0150 | 0.0145 | 0.0132 | 0.0153 | 0.0131 |
| | $T_W$ | 0.0715 | 0.0681 | 0.0545 | 0.0555 | 0.0541 | 0.0518 | 0.0523 | 0.0518 | 0.0518 | 0.0529 |
| Green Wealth | T | 0.1170 | 0.1180 | 0.1207 | 0.1201 | 0.1201 | 0.1155 | 0.1171 | 0.1314 | 0.1287 | 0.1366 |
| | $T_B$ | 0.0214 | 0.0239 | 0.0234 | 0.0252 | 0.0228 | 0.0226 | 0.0213 | 0.0257 | 0.0248 | 0.0286 |
| | $T_W$ | 0.0956 | 0.0941 | 0.0973 | 0.0949 | 0.0973 | 0.0929 | 0.0958 | 0.1057 | 0.1040 | 0.1081 |
| Green Growth | T | 0.1356 | 0.1299 | 0.1103 | 0.1013 | 0.0955 | 0.1015 | 0.0978 | 0.0980 | 0.1016 | 0.1089 |
| | $T_B$ | 0.0552 | 0.0600 | 0.0489 | 0.0469 | 0.0429 | 0.0468 | 0.0491 | 0.0499 | 0.0547 | 0.0597 |
| | $T_W$ | 0.0804 | 0.0699 | 0.0614 | 0.0544 | 0.0526 | 0.0547 | 0.0487 | 0.0480 | 0.0469 | 0.0492 |
| Green Benefits | T | 0.0326 | 0.0187 | 0.0224 | 0.0262 | 0.0189 | 0.0202 | 0.0247 | 0.0265 | 0.0147 | 0.0140 |
| | $T_B$ | 0.0056 | 0.0016 | 0.0018 | 0.0051 | 0.0035 | 0.0051 | 0.0029 | 0.0017 | 0.0015 | 0.0003 |
| | $T_W$ | 0.0271 | 0.0171 | 0.0206 | 0.0212 | 0.0155 | 0.0151 | 0.0218 | 0.0248 | 0.0132 | 0.0137 |

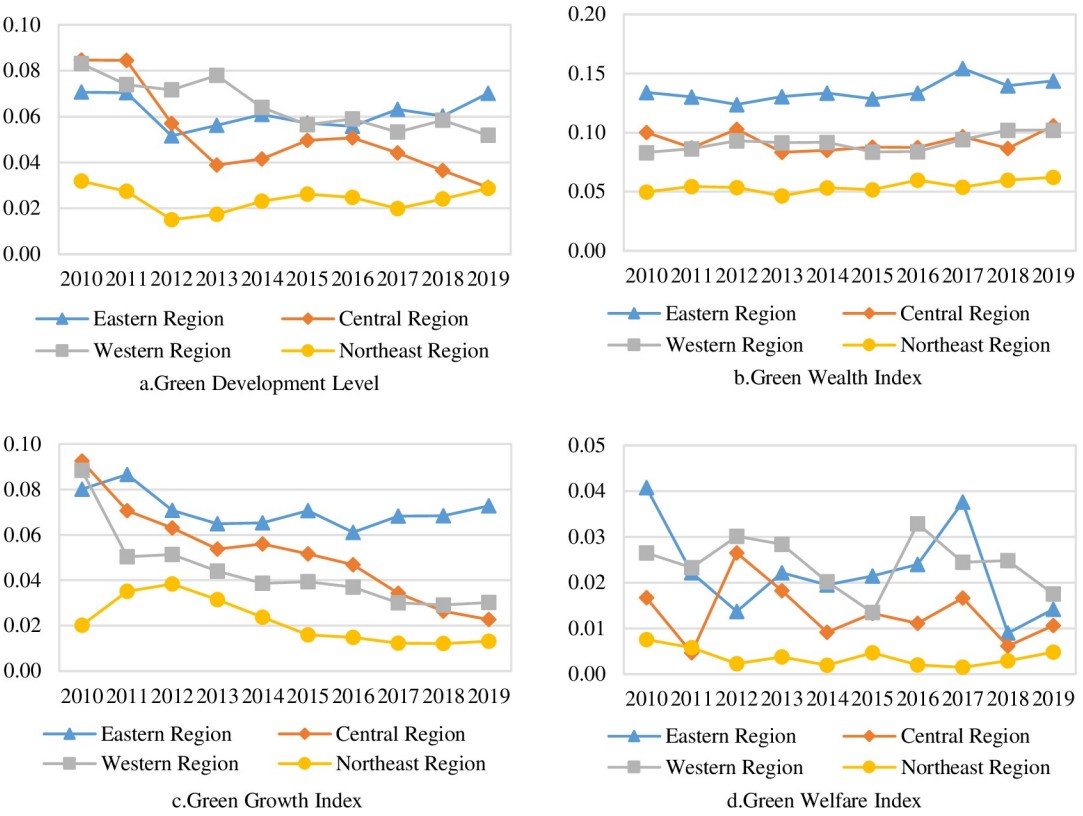

**Fig 5. Regional differences of China's green development level and its composition from 2010 to 2019.**

According to Table 2, the trends and main sources of the spatial differences in the national green development level and its composition index can be seen.

Firstly, the overall trend of the Thiel index of green development level is decreasing, which indicates that the overall differences among the four major regions are decreasing. Among them, the intra-group differences are larger than the inter-group differences, so the intra-group differences are the main source of the overall differences. As can be seen from Fig 5A, the intra-regional differences in both the central and western regions show a decreasing trend, while the intra-regional differences in the eastern and northeastern regions show a decreasing and then increasing trend.

Secondly, the Thiel indices of the green wealth index and green growth index is higher than that of green welfare index, indicating that the overall variation of the green wealth index and green growth index is the primary source of the overall variation of green development. The Thiel index of green wealth index shows a slight increase, indicating that the overall differences among the four regions have increased. Among them, the intra-group differences are larger than the inter-group differences, so the overall differences among the four regions mainly come from the intra-group differences. As can be seen from Fig 5B, the intra-regional differences in the east are significantly higher than those in the central, western and northeastern regions, with the intra-regional variation in the central region showing a smooth fluctuation and the other three regions showing a slight increasing trend.

Thirdly, the decreasing trend of the Thiel index of green growth index indicates that the overall differences among the four regions have slightly decreased. Among them, the within-group differences are similar to the between-group differences, so the within-group differences

and between-group differences are the main sources of overall differences. Among them, the between-group differences slightly increased during the examined period, and this widening of differences originated from the widening of differences between the eastern region and other regions (Fig 3C). As shown in Fig 5C, the intra-group differences in the four major regions show a decreasing trend.

Fourthly, the Thiel index of the green welfare index decreases to a lower state, indicating that the overall differences among the four major regions decrease significantly. Among them, the within-group differences are larger than the between-group differences, so the overall differences of the four regions mainly come from the within-group differences. From Fig 5D, it can be seen that the within-group differences of the four regions are reduced to a lesser extent.

A comprehensive analysis of the above shows that there are certain spatial differences in the national green development level and its composition index; the spatial differences in the green development level and green welfare index are narrowing, the spatial differences in the green wealth index are slightly expanding, and the spatial differences in the green growth index, intra-regional differences are narrowing while inter-regional differences are slightly expanding. Therefore, it is still imperative to encourage the central, western and northeastern regions with lower green growth index to find new green growth points in the green and low-carbon economic transformation, and provide more policy support to these regions to narrow the gap of green growth level between regions as much as possible.

## Spatial correlation analysis of China's green development level in the context of carbon neutrality

### Moran index

This paper uses the Moran index to measure the relevance or agglomeration of each province's green development level within the overall space of the country as a whole.

Firstly, it is the global spatial autocorrelation. Moran's $I$ is the global spatial autocorrelation index of China's green development level in specific years, $m$ is the number of provinces, $X_i$ and $X_s$ are the green development level values of provincial samples $i$ and $s$, respectively, $u$ is the average of provincial green development levels, and $W$ is the weight matrix of provincial spatial relationships, and this paper uses the inverse distance weight matrix.

$$Moran's\ I = (m\sum\nolimits_{i=1}^{m}\sum\nolimits_{s=1}^{m}W_{is}|X_i - u||X_s - u|)/[\sum\nolimits_{i=1}^{m}\sum\nolimits_{s=1}^{m}W_{is}\sum\nolimits_{i=1}^{m}(X_i - u)^2] \quad (10)$$

In the above equation, Moran's $I \in [-1, 1]$. if Moran's $I$ is positive, the green development level shows a positive spatial correlation; if Moran's $I$ is negative, the green development level shows a negative spatial correlation. Meanwhile, this paper uses the statistic $Z$ to test the significance of Moran's $I$ for the green development level of China in each year. E($I$) and VAR($I$) are the mean and variance of Moran's $I$, respectively.

$$Z = [I - E(I)]/\sqrt{VAR(I)} \quad (11)$$

If $Z>0$ and $P\leq0.1$, the provincial level of green development in China is significantly positively correlated; if $Z<0$ and $P\leq0.1$, the provincial level of green development in China is significantly negatively correlated; if $P>0.1$, Moran's $I$ and correlation of green development level are not significant.

Secondly, it is the local spatial autocorrelation index. $I_i$ is the local spatial autocorrelation index. If $I_i>0$, it means the province is in the "high-high" or "low-low" cluster of green development level, if $I_i<0$, it means the province is in the "low-high" or "high-low" cluster of green

Table 3. The overall autocorrelation of China's green-development level and its composition index from 2010 to 2019.

| Type | | 2010 | 2011 | 2012 | 2013 | 2014 | 2015 | 2016 | 2017 | 2018 | 2019 |
|---|---|---|---|---|---|---|---|---|---|---|---|
| Green Development | Moran's I | 0.0089 | 0.0016 | 0.0029 | 0.0144 | 0.0141 | 0.0195 | 0.0200 | 0.0067 | 0.0098 | 0.0088 |
| | Z | 1.2185 | 1.0145 | 1.0463 | 1.3758 | 1.3651 | 1.5180 | 1.5373 | 1.1964 | 1.2722 | 1.2460 |
| | P | 0.2230 | 0.3103 | 0.2954 | 0.1689 | 0.1722 | 0.1290 | 0.1242 | 0.2315 | 0.2033 | 0.2128 |
| Green Wealth | Moran's I | 0.0575 | 0.0459 | 0.0465 | 0.0434 | 0.0424 | 0.0514 | 0.0524 | 0.0419 | 0.0256 | 0.0350 |
| | Z | 2.5656 | 2.2568 | 2.2716 | 2.1851 | 2.1644 | 2.4211 | 2.4459 | 2.1479 | 1.6938 | 1.9646 |
| | P | 0.0103 | 0.0240 | 0.0231 | 0.0289 | 0.0304 | 0.0155 | 0.0145 | 0.0317 | 0.0903 | 0.0495 |
| Green Growth | Moran's I | 0.0120 | 0.0194 | 0.0279 | 0.0292 | 0.0371 | 0.0484 | 0.0540 | 0.0455 | 0.0517 | 0.0499 |
| | Z | 1.3181 | 1.5391 | 1.7582 | 1.7928 | 2.0208 | 2.3472 | 2.5029 | 2.2733 | 2.4432 | 2.3922 |
| | P | 0.1875 | 0.1238 | 0.0787 | 0.0730 | 0.0433 | 0.0189 | 0.0123 | 0.0230 | 0.0146 | 0.0167 |
| Green Benefits | Moran's I | 0.1717 | 0.0601 | 0.1257 | 0.1580 | 0.1079 | 0.1923 | 0.0910 | 0.1398 | 0.0043 | 0.0868 |
| | Z | 5.7065 | 2.6393 | 4.4958 | 5.2848 | 4.0091 | 6.2804 | 3.5131 | 4.8171 | 1.0648 | 3.3388 |
| | P | 0.0000 | 0.0083 | 0.0000 | 0.0000 | 0.0001 | 0.0000 | 0.0004 | 0.0000 | 0.2870 | 0.0008 |

development level.

$$I_i = [(X_i - u)/S] \cdot [\sum_{s=1}^{m} W_{is}(X_s - u)] \qquad (12)$$

## Spatial correlation analysis

To understand the overall spatial clustering and dispersion degree of China's green development level from 2010 to 2019, the analysis is developed through the calculation results of the global Moran's index Eq (10) to (11), as shown in Table 3. As can be seen from the table, Moran's *I* of green development level did not pass the significance test during the period under examination, and there was no significant spatial dependence of green development level in each province during the period under examination; while Moran's *I* of green wealth index, green growth index and green welfare index performed significantly in the vast majority of years. When Moran's *I* is positive, it means that the index has spatial agglomeration, in other words, areas with high index are adjacent to other areas with high index, and geographical units with low index are adjacent to other geographical units with low index; if Moran's *I* is negative, it means that the index is spatially divergent, in other words, areas with high index are adjacent to areas with low index. Specifically, the Moran's *I* values of both the green wealth index and the green welfare index are low and generally show a decreasing trend, indicating that the respective spatial agglomeration of these two index is generally weak; the Moran's *I* value of the green growth index generally shows an increasing trend from 0.0120 to 0.0499, indicating that the spatial clustering of the green growth index is weak but the clustering trend is increasing. Therefore, this paper argues that provinces in different regions should give full play to their relative advantages according to local conditions, establish a suitable local green development model, create a cross-provincial green development cooperation circle, and reduce the probability of polarization.

The local Moran's *I* index can effectively describe the degree of agglomeration and dispersion of a region and adjacent regions within a local area, and thus this paper is used to reveal the local spatial divergence characteristics of the green development level and its composition index in Chinese provinces from 2010–2019. Based on the results of the calculation of the local Moran index Eq (12) of green development level, the green development level can be divided into the following four evolutionary patterns (Table 4), and their composition index are

**Table 4. Local spatial clustering table of China's green development level from 2010 to 2019.**

|  | HH | LH | LL | HL |
|---|---|---|---|---|
| 2010 | Liaoning, Jilin, Heilongjiang, Zhejiang, Fujian | Anhui, Gansu | Tianjin, Hebei, Shanxi, Shandong, Henan, Hubei, Hunan, Guangxi, Guizhou, Chongqing, Shaanxi, Ningxia, Xinjiang | Beijing, Inner Mongolia, Shanghai, Jiangsu, Jiangxi, Guangdong, Hainan, Sichuan, Yunnan, Qinghai |
| 2015 | Jilin, Heilongjiang, Shanghai, Jiangsu, Zhejiang, Fujian | Liaoning, Anhui, Jiangxi, Hainan | Hebei, Shanxi, Shandong, Henan, Hubei, Guangxi, Gansu, Chongqing, Guizhou, Shaanxi, Ningxia, Xinjiang | Beijing, Tianjin, Inner Mongolia, Hunan, Guangdong, Sichuan, Yunnan, Qinghai |
| 2019 | Jilin, Shanghai, Jiangsu, Zhejiang, Fujian | Tianjin, Anhui, Liaoning, Gansu, Jiangxi, Guangxi | Hebei, Shanxi, Shandong, Henan, Hubei, Hunan, Hainan, Chongqing, Guizhou, Shaanxi, Ningxia, Xinjiang | Beijing, Inner Mongolia, Heilongjiang, Guangdong, Sichuan, Yunnan, Qinghai |

interpreted in the same way: firstly, it is the high-high (HH) agglomeration area, which means that these provinces show a high level of spatial agglomeration effect of green development with neighboring provinces; secondly, it is the low-high (LH) agglomeration area, which is the transition area where the province's low level of green development changes to the high level of neighboring provinces; thirdly, it is the low-low (LL) agglomeration area, which means that these provinces exhibit a low level of spatial agglomeration effect of green development with neighboring provinces, which manifests itself as a collapse pattern; and fourthly, it is high-low (HL) agglomeration, which is a polarization pattern where the province has a high level of green development while the neighboring provinces have a low level of green development. In 2019, China's green development level agglomeration area mainly belong to LL and HL agglomeration areas, where LL agglomeration areas are mainly distributed in the central and western regions, and HL agglomerations are mainly in Beijing and Guangdong in the East and Inner Mongolia, Sichuan, Yunnan, and Qinghai in the West. Meanwhile, HH agglomerations mainly fall in the developed eastern coastal areas. The above shows that there is a certain positive spatial matching relationship between the spatial agglomeration of green development level and the spatial agglomeration of green wealth index and green growth index (combined with Fig 4), which does not form a close correlation with the spatial agglomeration of green welfare index. In addition, there are 10 provinces where spatial agglomerations jumped from 2010 to 2019, indicating that the spatial pattern of green development level in China is relatively stable.

The local Moran index scatter plot of each composition index of green development was calculated by using Stata software (Fig 6). From the spatial agglomeration areas of each component index of green development, the agglomeration areas of the green wealth index are mainly in the LL and HL agglomeration areas, the agglomeration areas of the green growth index are mainly in the LL, HH and LH agglomeration areas, and the agglomeration areas of the green welfare index are mainly in the LL agglomeration area. (1) In 2019, the HH agglomeration areas of the green wealth index are distributed in four provinces, namely, Liaoning, Jilin, Heilongjiang, and Fujian, and the LL agglomerations areas are distributed in 13 provinces, namely, Beijing, Tianjin, Hebei, Shandong, Shanxi, Anhui, Henan, Hubei, Chongqing, Guizhou, Shaanxi, Ningxia, and Xinjiang, and the spatial agglomerations have leaped in five provinces; (2) In 2019, the HH agglomeration areas of the green growth index are distributed in seven provinces, namely Tianjin, Shanghai, Jiangsu, Zhejiang, Fujian, Hubei and Hainan, and the LL agglomeration areas are distributed in 11 provinces, namely Shanxi, Inner Mongolia, Liaoning, Jilin, Heilongjiang, Chongqing, Yunnan, Shaanxi, Gansu, Ningxia, and Xinjiang, with seven provinces in which spatial agglomerations have jumped; (3) In 2019, the HH agglomeration areas of the green welfare index are distributed in 11 provinces, namely Jilin, Heilongjiang, Beijing, Hebei, Jiangsu, Shandong, Shanxi, Henan, Anhui, Inner Mongolia, and

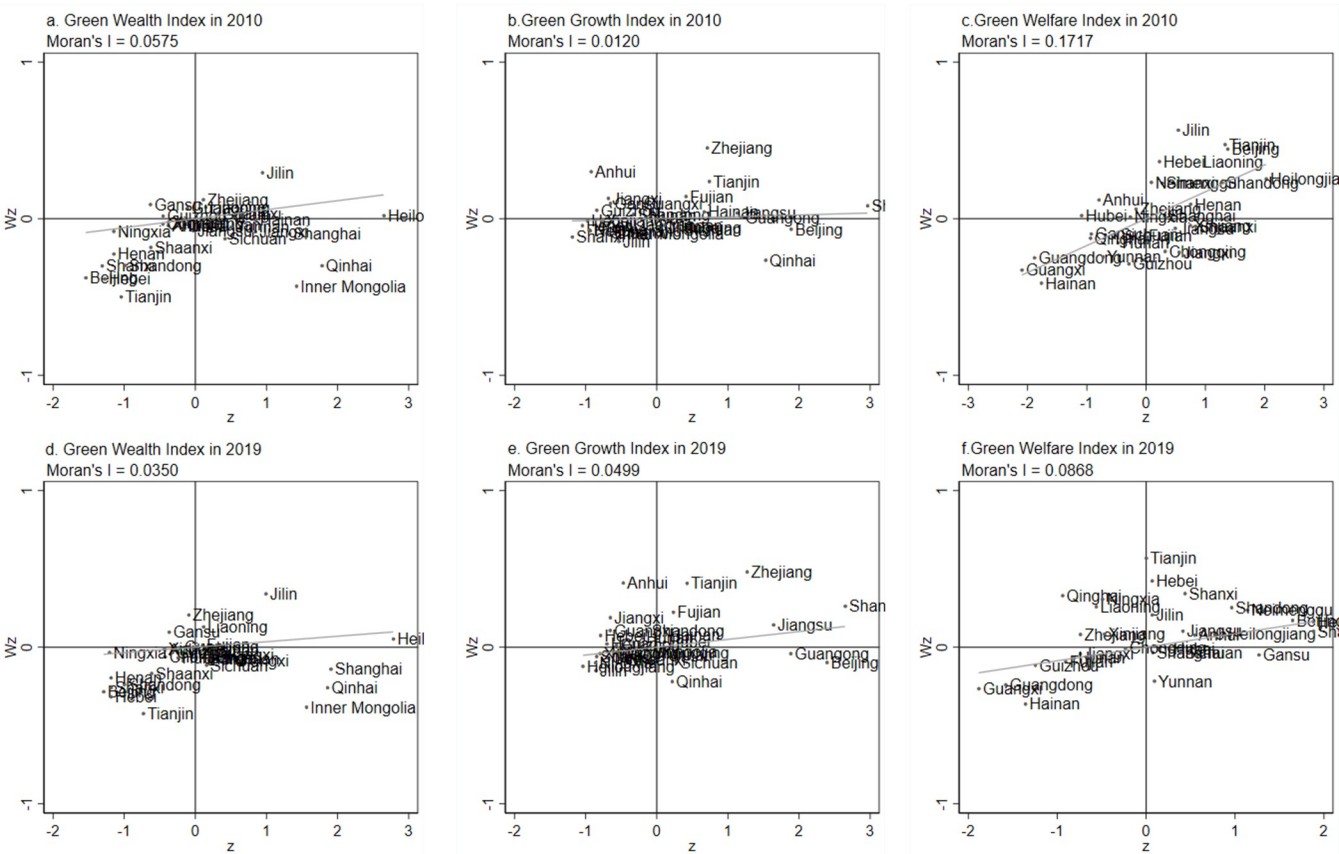

**Fig 6. Local Moran's I index scatter plot of China's green-development component index from 2000 to 2019.**

Shaanxi, and the LL agglomeration areas are distributed in 7 provinces, namely Fujian, Guangdong, Hainan, Jiangxi, Hunan, Guangxi, and Guizhou, with 14 provinces where the spatial agglomerations have jumped. Comprehensive analysis of the above reveals that the LL agglomerations areas of green wealth and the HH agglomeration areas of green growth are mainly concentrated in the eastern region, reflecting to a certain extent that green development in the eastern region is mainly promoted through green growth, but is constrained by the low green wealth index; the LL agglomeration areas of green wealth and the LL agglomeration areas of green growth both are mainly located in the western region, indicating that green development in some western provinces has more ecological and economic difficulties.

## Discussion

This study and the studies of Cheng et al. [29] and Xu et al. [25] both found that: firstly, the green development level and its component indices in China are both low and show an upward trend, with certain spatial differences which show a relatively decreasing trend; secondly, the average annual growth rate of green development level in China calculated in this study is 5.05%, which is similar to 3.77% obtained by Cheng et al. [29]; thirdly, there is a distinct spatial clustering phenomenon of green wealth index, green growth index and green welfare index.

The results of this study also differ from the findings of existing studies. Specifically, firstly, this study finds that green development shows a spatial distribution pattern of

Northeast > East > West > Central, while Cheng et al. and Xu et al. argue that eastern China has the highest level of green development while northeast China has the second highest level [25, 29]; secondly, this study finds that eastern region has the fastest growth rate of green development level, while Xu et al. argue that the less developed central and western provinces have faster growth in green development level [25]; third, this study finds that there is no significant spatial clustering of green development level in China as a whole, while Cheng et al. argue that the spatial clustering is distinct [29].

The reasons for the differences between some results obtained in this study and other studies are: firstly, the green wealth indicators in this study involves ecological service functions, while the green wealth indicators designed by Cheng et al. involves environmental stress, and the rest of the green wealth indexes are the same in both studies [29]. The green wealth indicators designed in this study highlights the connotation of ecosystems actively benefiting humans from ecological service functions [12], and thus Northeast has a clear advantage in total natural resources and density, and has elevated its own green development with a very high green wealth index. However, starting from 2018, green development level in East surpasses that of Northeast. Secondly, the indicator system of this study has a relatively even number of green development indicators in ecological, economic, and social aspects, while the indicator system of Xu et al. is more focused on natural resource use, environmental quality, environmental governance, and ecological protection, and includes fewer indicators of growth quality and green life [25], which is the reason why the regions with fast growth rates of green development in this study are different from them. In addition, from the perspective of green growth, which has the largest gap between green development in East and Midwest, the green growth indicators in this study emphasize more on reflecting regional advantages in green production, green technology and new energy, thus the growth rate of green growth in East is faster. However, Central and Western are relatively weak in green technology innovation capability and are taking over the traditional industry transfer from East, which is not conducive to their own ecological economic quality improvement [51, 52], and they do not have advantages in the green industry field with green technology innovation as the core [53–55], so the green growth in Central and Western is relatively slow. Thirdly, because this study included indicators not covered in the study of Cheng et al., such as ecological service function, green industry, and carbon emissions, the advantages of green development reflected in each region are different, and the relatively smaller differences in green development levels among regions allow China to show no overall spatial clustering of green development levels. In addition, this study used the entropy method to measure green development level, while Cheng et al. used the projection pursuit evaluation model [29], which is the reason for some differences in the conclusions of the two studies.

Due to the richness and complexity of the connotation and extension of green development theory and the limited statistical information at present, individual areas of green development practice such as green building promotion [56] and CCUS application [57] have not been included in the indicator system. Building an indicator system to reflect regional green development practices still faces great challenges, and the following aspects will be strengthened in the future: first, continue to carry out research work on green development practices and revise the green development indicator system with regional characteristics; second, strengthen the research on the influence mechanism of regional green development, and deeply analyze the heterogeneity, driving factors and key paths of green development in different regions.

## Conclusions and recommendations

Based on this major background of carbon neutrality, the green development level indicator system is constructed according to the theoretical framework of "three layers" of green

development, and the entropy value method, Thiel index and Moran index are applied to measure the green development level of 30 Chinese provinces from 2010 to 2019 and analyze the spatial differentiation, and the following conclusions are obtained.

Firstly, from the perspective of temporal change, China's green development level and its composition index are on the rise with an average annual growth rate of 5.05%, among which the green growth index has a larger growth rate while the green wealth index and green welfare index have a smaller growth rate; from the perspective of spatial evolution, China's green development shows a spatial distribution pattern of Northeast > East > West > Central, and the eastern region has the fastest growth rate. The Northeast, which has the highest green wealth index, has not transformed its natural resource advantages into more economic value instead.

Secondly, there are certain spatial differences in both the national green development level and its composition index; the spatial differences in green development level and green welfare index and green growth index are gradually decreasing, while the spatial differences in green wealth index and inter-regional differences in green growth index are slightly expanding.

Thirdly, there is no obvious spatial agglomeration phenomenon in the national green development level as a whole, and the local HH agglomeration areas in 2019 are mainly located in the developed eastern coastal regions, and the local LL agglomeration areas are mainly distributed in the western regions. Green wealth index, green growth index and green welfare index have distinct spatial clustering phenomenon. The spatial agglomeration of green development level has a positive spatial matching relationship with green wealth index and green growth index, but does not form a close correlation with green welfare index. Green development in the eastern region is mainly driven by green growth, while green development in some western provinces has more difficulties in ecological and economic aspects.

Based on the above conclusions, the future promotion of green development should be based on stimulating the endogenous power of green development and narrowing the regional differences of green growth index. This paper makes the following recommendations.

Firstly, the government should fully activate the key elements of talent, natural resources, capital and technology. Talent is the key to green development, and governments at all levels should stimulate the vitality of all fields of green development, and should actively cultivate and introduce green technological innovation and application talents, and introduce operational talents in the field of green industries and operators of natural resources development and operation; provide more opportunities and platforms for these talents to participate through project planning and construction, and the establishment of incentive funds and entrepreneurship funds. Natural resources are the basis of green development, activating natural resource elements, and should actively explore the path and mechanism for realizing the value of ecological products in line with the actual situation of the region, while further developing renewable energy, focusing on the development of photovoltaic and wind energy on a scale of efficiency. Capital is the blood of green development, and provinces should improve the multi-level capital market, actively use green finance, and establish social capital incentive mechanisms for remote areas. Technology is the breakthrough of green development, all regions should actively explore the ecological protection and restoration technology of landscape, forest, field, lake and grass, accelerate the introduction and research and development of green environmental protection technology in carbon reduction, pollution reduction and recycling, and set up special support policies and supporting funds to stimulate the innovation and application of these technologies.

Secondly, the government tries to narrow the green growth gap between the central, western and northeastern regions and the eastern region. Improving the green growth index in the central, western and northeastern regions is the key to narrowing the inter-regional differences

between green growth. Governments at all levels should adhere to the principle of localization, give full play to their relative advantages, actively search for new green economic growth points, establish a green growth model suitable for local areas, create a cross-provincial green development cooperation circle, and promote the coordinated development of regional green economy. Provinces with natural resource endowment should focus on resource characteristics, transform natural resource advantages into product advantages or even industrial advantages, and finally transform more economic values; provinces with better green industry foundation should find gaps with developed regions, then make up short boards, strengthen weaknesses, make green industries more refined, more detailed and stronger, continuously improve green industry supporting services, and continue to extend green industry chains.

## Supporting information

**S1 File. Corresponding data of Figs 2~6.**
(DOCX)

**S1 Data.**
(ZIP)

## Acknowledgments

The authors appreciate the efforts and constructive feedback of the editors and the reviewers, which helped improve the manuscript.

## Author Contributions

**Conceptualization:** Kun Liang.

**Data curation:** Kun Liang, Li Luo.

**Formal analysis:** Kun Liang.

**Funding acquisition:** Li Luo.

**Methodology:** Kun Liang.

**Software:** Kun Liang.

**Writing – original draft:** Kun Liang, Li Luo.

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
