## [Decision Letter · Decision Letter 0]

20 Feb 2023

PONE-D-23-00550Measurement of China’s Green Development Level and Its Spatial Differentiation in the Context of Carbon NeutralityPLOS ONE

Dear Dr. Liang,

Thank you for submitting your manuscript to PLOS ONE. After careful consideration, we feel that it has merit but does not fully meet PLOS ONE’s publication criteria as it currently stands. Therefore, we invite you to submit a revised version of the manuscript that addresses the points raised during the review process.

We look forward to receiving your revised manuscript.

Kind regards,

László Vasa, PhD

Academic Editor

PLOS ONE

Journal Requirements:

3. PLOS requires an ORCID iD for the corresponding author in Editorial Manager on papers submitted after December 6th, 2016. Please ensure that you have an ORCID iD and that it is validated in Editorial Manager. To do this, go to ‘Update my Information’ (in the upper left-hand corner of the main menu), and click on the Fetch/Validate link next to the ORCID field. This will take you to the ORCID site and allow you to create a new iD or authenticate a pre-existing iD in Editorial Manager. Please see the following video for instructions on linking an ORCID iD to your Editorial Manager account: https://www.youtube.com/watch?v=_xcclfuvtxQ.

Reviewers' comments:

Reviewer's Responses to Questions

**Comments to the Author**

1. Is the manuscript technically sound, and do the data support the conclusions?

Reviewer #1: Yes

Reviewer #2: Yes

2. Has the statistical analysis been performed appropriately and rigorously? 

Reviewer #1: Yes

Reviewer #2: Yes

3. Have the authors made all data underlying the findings in their manuscript fully available?

Reviewer #1: No

Reviewer #2: Yes

4. Is the manuscript presented in an intelligible fashion and written in standard English?

Reviewer #1: Yes

Reviewer #2: Yes

5. Review Comments to the Author

Reviewer #1: The article addresses a very interesting topic. The article is well structured, the ideas are presented in a logical, concise order. The statements are supported by specific data, that are analysed in a proper way by the authors.

The article has the potential to be published, which is why I recommend some revisions

1. in the introduction, the authors must also mention the efforts made by public authorities in other countries in the transition process towards low carbon economy and green development. In this sense, the experience of the European Union countries in the field of green transition must be presented in the introduction. (https://climate.ec.europa.eu/eu-action/european-green-deal_en)

2. The discussion section must be expanded and include the conclusions of similar studies that confirm or not the results obtained by the authors

3. The paper contributes to existing research, and it is interesting and relatively new of its kind, however the limitations and future research should be enriched.

4. The authors must extend the references list that can be useful for different parts of the manuscript (introduction, discussions etc.).

a) Adams, B. (2019). Green development: Environment and sustainability in a developing world. Routledge.

b) Han, M. S., Yuan, Q., Fahad, S., & Ma, T. (2022). Dynamic evaluation of green development level of ASEAN region and its spatio-temporal patterns. Journal of Cleaner Production, 362, 132402.

c) Wu, H., Li, Y., Hao, Y., Ren, S., & Zhang, P. (2020). Environmental decentralization, local government competition, and regional green development: Evidence from China. Science of the total environment, 708, 135085.

Reviewer #2: The study deals with a very current topic. The colleagues processed the issue in a very high-quality way. With careful research work, with a really detailed explanation. The authors have also carefully selected the appropriate mdsertan and their summary findings are exemplary for representatives of other states as well.

6. PLOS authors have the option to publish the peer review history of their article (what does this mean?). If published, this will include your full peer review and any attached files.

Reviewer #1: No

Reviewer #2: **Yes: **Prof. Dr. Boros Anita

---

## [Author Response · Author response to Decision Letter 0]

14 Mar 2023

Dear Reviewers.

 I am very grateful to the two reviewers for their constructive comments, and I am doubly honored. In response to your comments, I have revised this paper accordingly.

 First, about the data used in the submission of this paper. I have organized all the data in the indicator system of this paper in a master table, which also contains the calculation results of the green development level and its component indices as well as pictures. The raw data involved in this paper are organized in separate tables and categorized in different files. Moreover, the links to the data sources have been updated in this paper so that readers can use them directly. In order to increase the openness of the calculation methods of the indicators, the calculation methods and corresponding indicators of Green Total Factor Productivity (GTFP) are included in the notes at the bottom of Table 1. Considering China's accession to WTO in 2003, the GDP data used to calculate GTFP are deflated with 2004 as the base period, and the amount of fixed asset investment is deflated with 2004 as the base period for the price index. It should be noted here that the same method of deflating other years is also desirable.

 Second, regarding the need for the introduction to describe the efforts of public authorities in other countries in the transition to a low-carbon economy and green development. This paper briefly mentions the carbon reduction practices in the United States, Japan, and Korea in the introduction, and introduces the carbon reduction instruments and experiences proposed by the European Union and its published European Green Deal to enhance the necessity and importance of choosing carbon neutrality as the context for this study.

 Third, about adding the discussion section. I have added a new section, Discussion, specifically to this paper. This section first selected two similar studies published in recent years with Chinese provinces as samples and compared the indicator system and conclusions from these two articles with the results of my study. In terms of the main conclusions, my study is identical or similar to the conclusions of these two studies. In terms of some of the more detailed conclusions, there are major or minor differences between these studies, which are due to the differences in the indicators these studies included in the green development level indicator system and the different methods used to measure the green development level. At the end of the discussion section, I still pointed out the limitations of this study and the direction of future research.

 Fourth, about increasing the references cited in various parts of the paper. In order to enhance the persuasive power of the ideas and academic standardization of this paper, the introduction and discussion of this paper further increase the citations in related studies, especially in the discussion section of this paper also cites a lot of other related studies as the supporting evidence of this paper's view. In addition, the original manuscript involved a lot of Chinese literature, which has been partially replaced with similar literature from international journals and has been specifically marked out in this revision.

Thank you and best regards.

Yours sincerely,

Dr. Kun LIANG

March 12, 2023

---

## [Editor Report · Decision Letter 1]

27 Mar 2023

Measurement of China’s Green Development Level and Its Spatial Differentiation in the Context of Carbon Neutrality

PONE-D-23-00550R1

Dear Dr. Liang,

We’re pleased to inform you that your manuscript has been judged scientifically suitable for publication and will be formally accepted for publication once it meets all outstanding technical requirements.

Kind regards,

László Vasa, PhD

Academic Editor

PLOS ONE
---

## [Editor Report · Acceptance letter]

30 Mar 2023

PONE-D-23-00550R1 

Measurement of China’s Green Development Level and Its Spatial Differentiation in the Context of Carbon Neutrality 

Dear Dr. Liang:

I'm pleased to inform you that your manuscript has been deemed suitable for publication in PLOS ONE. Congratulations! Your manuscript is now with our production department. 

Kind regards, 

on behalf of

Prof. Dr. László Vasa 

Academic Editor

PLOS ONE